# Study on the Influence of Reinforced Particles Spatial Arrangement on the Neutron Shielding Performance of the Composites

**DOI:** 10.3390/ma15124266

**Published:** 2022-06-16

**Authors:** Weiqiang Sun, Guang Hu, Hu Xu, Yanfei Li, Chao Wang, Tingxuan Men, Fu Ji, Wanji Lao, Bo Yu, Liang Sheng, Jinhong Li, Qinggang Jia, Songqi Xiong, Huasi Hu

**Affiliations:** 1School of Nuclear Science and Technology, Xi’an Jiaotong University, Xi’an 710049, China; sunweiqiang@stu.xjtu.edu.cn (W.S.); xuhu_xu@stu.xjtu.edu.cn (H.X.); liyanfei@xjtu.edu.cn (Y.L.); wang_chao@stu.xjtu.edu.cn (C.W.); 18161768335@163.com (T.M.); zndxjf05822@163.com (F.J.); nxlao132@stu.xjtu.edu.cn (W.L.); 2State Key Laboratory of Light Alloy Foundry Technology for High-End Equipment Shenyang Research Institute of Foundry Co., Ltd., Shenyang 110022, China; yub@chinasrif.com; 3State Key Laboratory of Intense Pulsed Radiation Simulation and Effect, Northwest Institute of Nuclear Technology, Xi’an 710024, China; liangs@nist.ac.cn; 4Institute of Applied Physics and Computational Mathematics, Beijing 100094, China; jinh@163.com (J.L.); gqjia_xjtu@163.com (Q.J.); 5Nuclear and Radiation Safety Supervision Station of Shaanxi Province, Xi’an 710054, China; songqix@163.com

**Keywords:** neutron shielding materials, particle spatial arrangement, random model, MCNP code

## Abstract

Particle-reinforced composites are widely applied as nuclear radiation shielding materials for their excellent comprehensive properties. The work aimed to calculate the influence of the functional reinforced particles spatial arrangement on the neutron shielding performance of composites and attempted to explain the influence mechanism by investigating the neutron flux distribution in the materials. Firstly, four suitable physical models were established based on the Monte Carlo Particle Transport Program (MCNP) and mathematical software MATLAB, namely the RSA (Random Sequential Adsorption) Model with particles random arrangement and FCC Model, BCC Model and Staggered Arrangement Model (SA Model) with particle periodic arrangements. Later, based on these four physical models, the neutron transmittance of two kinds of typical B_4_C reinforced composites, 316 stainless steel matrix composite and polyethylene matrix composite, were calculated under different energy neutrons sources (0.0253 eV, 50 eV, 50 keV, fission spectrum, ^241^Am-Be spectrum and 14.1 MeV) and the neutron flux distribution in the 316 stainless steel composite was also analyzed under 0.0253 eV neutron and fission neutron sources. The results indicated that the spatial arrangement of B_4_C has an impact on the neutrons shielding performance of the composite and the influence changes with neutron energy and B_4_C content. It can be concluded that the RSA model and the periodic arrangement models can be used in different calculation cases in the future.

## 1. Introduction

Nuclear radiation shielding materials with functional and structural integration can adapt to more severe service environments and they are of great significance to ensure the normal operation of increasingly abundant nuclear facilities [1]. On the whole, the particle reinforced composite is a type of material with both excellent shielding performance and mechanical property, and they are widely used as shielding materials [2]. For example, borated stainless steel and boron-added aluminum alloys are extensively used in storage and transportation containers for spent nuclear fuel because of their excellent thermal neutron absorption performance and good mechanical properties [3,4,5]. At the same time, polymers are also used in spent storage casks to shield neutrons because the microscopic cross section of elastic collision between hydrogen element and neutron is large [5,6]. Common matrix materials for neutron shielding include 316 stainless steel and polyethylene (PE) [7,8,9,10] and a B_4_C particle is used as neutron absorption material because of its advantages of high thermal neutron capture cross section, high strength, low density and inactive chemical properties [11]. The fundamental reason is that the inelastic scattering cross sections of heavy elements relative to neutrons are relatively high, such as iron, chromium and nickel in 316 stainless steel, and the elastic scattering cross sections of light elements relative to neutrons are relatively high, such as carbon and hydrogen in PE, and the absorption cross sections of ^10^B in B_4_C are extremely high. Therefore, in this paper, we focused our attention on B_4_C-reinforced 316 stainless steel composites and polyethylene composites. 

When designing neutron shielding materials, the shielding performance of the designed materials should be evaluated by simulation calculation. At present, particle transport calculation based on the Monte Carlo method is still the main method, and the MCNP program is one of the most common such programs. Traditionally, a Uniform Filling Model (UFM) was established to calculate the radiation shielding performance of particles reinforced composites [12,13,14,15,16]. Moreover, there is a conventional hypothesis that the matrix material and reinforced particles are uniformly filled in the shield according to the composition ratio in the UFM, but the parameters of particles, such as the size, the shape and the shape parameters, have not been taken into consideration in details. These types of models and hypotheses are only applicable to the case of very small size reinforced particles. 

For the accurate calculation of the shielding property, the models considering enhanced particle structure are becoming a novel orientation. Some models involving particle size and particle shape parameters have also been developed in recent years. As far as investigation is concerned, some researchers built a periodic arrangement model of reinforced particles to calculate the influence of particle size on the shielding performance; e.g., Jaewoo Kim successfully established a model in which a spherical particle is located at the center of a cubic, and he drew the conclusion that nano-B_4_C/HDPE (300 nm) had a better thermal neutron shielding performance than micro-B_4_C/HDPE (300 μm). This model was also applied in Hoda Alavian’s work to study how tungsten’s size influences the shielding property of LDPE/W composites [17,18]. Shuo Chen prepared the hollow spherical metal foam materials with 2.0–5.2 mm hollow spheres and set up three periodic models to compare the shielding performances of the materials with different size foams [19,20]. In these periodic arrangement models, they assumed that reinforced particles were arranged periodically. 

However, the problem is that reinforcement particles are usually randomly distributed in the matrix material in the process of material preparation and the above two types of models cannot match simulations and experiments well. In order to make the design more reliable, our group has constructed an RSA Model and GRM Model in which the particles are randomly filled in the matrix in order to study how the particle’s size, the shape and shape parameters influence the radiation shielding property of the composites [21]. We have also manufactured 1.53 mm spherical B_4_C-reinforced 304 stainless steel composites in earlier studies [22].

Another point to question is that difference between simulation results and experiment results is very small in our research study, while in Shuo Chen’s research study, it can be seen that the difference between simulation results and experiment results is large [19,20]. Therefore, there is an adequate reason to suspect that enhanced particle spatial arrangement has an effect on the shielding properties of composites. At the same time, the neutron energies between our research and Shuo Chen’s research are different; thus, neutron energy may also affect the simulation results between random model and periodic models. In addition, it has been concluded from our previous research that the modeling time of random model is much longer than the periodic models, although it is more accurate for shielding calculation, and the modeling time of RSA Model increases rapidly with the increase in particle volume fraction [21]. Therefore, it is of great significance to study how the spatial arrangement of particles influences the shielding performance of the composites and when the periodic model can be used to replace the random model to improve computational efficiency. However, there are few research studies on this aspect reported.

For the foregoing reasons, in this paper, one random model, RSA Model, and three periodic models, FCC Model, BCC Model and SA Model, were established based on the Monte Carlo Particle Transport Program—MCNP (MCNP5) [23] and the mathematical software—MATLAB (2016a) [24,25]—and the neutron transmittances of two kinds of typical B_4_C reinforced composites were calculated under different energy neutrons (0.0253 eV, 50 eV, 50 keV, fission spectrum, ^241^Am-Be spectrum and 14.1 MeV). Among these neutron energies, 0.0253 eV is the thermal neutron, 14.1 MeV is the neutron energy of D-T fusion, the fission spectrum is the most common neutron spectrum, the ^241^Am-Be spectrum will be compared with the experiment results and 50 eV and 50 keV represent resonance neutron and intermediate neutron, respectively. In addition, for 0.0253 eV neutrons and fission neutrons, the neutron flux distributions at different depths of the 316 stainless steel composites were also obtained and analyzed. The mechanism of the influence of particle spatial arrangement on the neutron shielding performance of the composite was explained by coming up with ‘alternating shielding’ and ‘neutron channels’ and combining with the microscopic cross section.

## 2. Materials and Methods

### 2.1. Materials

In this paper, B_4_C-reinforced 316 stainless steel composites and PE composites were researched because of their extensive applications in neutrons shielding. The components ratio and density of the 316 stainless steel, PE, B_4_C particle and the composites used in this paper are shown in Table 1. 

In the previous study, our group completed the optimization design of a 304 stainless steel matrix composite and the polyethylene composite with high B_4_C content. In the former composite, the mass fraction of B_4_C can reach 24.68 wt% and the volume fraction is about 50.67 vol.% [22]. The mass fraction of B_4_C in the latter composite is 45.11 wt% and the volume fraction is about 23.65 vol.% [26]. Therefore, in this study, we considered that the volume fraction of B_4_C in two types of composites are 1 vol.%–30 vol.%. Moreover, we assumed that spherical B_4_C can be perfectly combine with the matrix material and that there is no gap between them.

The neutron microscopic total cross sections (barns) of main nuclides were obtained from databases [27] and they are listed in Table 2.

### 2.2. Modeling Methods of Four Models

Four different physical models, RSA Model, FCC Model, BCC Model and SA Model, were established based on the MCNP5 code and MATLAB (R2016a). Therein, B_4_C particles were filled randomly in RSA Model and they are filled periodically in the FCC Model, BCC Model and SA Model.

#### 2.2.1. RSA Model

The establishment of RSA Model by MATLAB software and MCNP program can be divided into two steps: (1) According to RSA (Random Sequential Adsorption) means [28], the position parameters of random particles were generated by MATLAB program; (2) the input file was written by MATLAB and run by MCNP program. The specific implementation method of the generation of position parameters is that the first geometric center point was randomly generated in the considered volume and then the next point was created in the remaining volume, and the points were produced circularly until the volume fraction occupied by particles reached the set value. More detailed modeling processes and the specific validation of this model have been performed in previous studies [21]. The maximum volume fraction that can be achieved theoretically was 38 vol.%. The RSA Model is shown in Figure 1.

#### 2.2.2. BCC Model and FCC Model

The BCC Model was body centered cubic model, and the FCC Model was face centered cubic model. They were constructed by referring to metal lattice arrangements. The U CARD and FILL CARD in MCNP5 code were used to establish the periodic repetition unit, and MATLAB was applied to write the input files for MCNP. The BCC Model unit and FCC Model unit were first set up as the filling body, and then the units were filled in the entire shield. The volume fraction of the B_4_C particle was realized by controlling the side length of the cubic unit. The unit side lengths of BCC Model and FCC Model were calculated according to the following formulas:(1)lBCC=(((1+8×18)×43πr3)/per)13
(2)lFCC=(((8×18+6×12)×43πr3)/per)13
where r is the radius of spherical particles; per is the volume fraction of particles.

In fact, there are two effective spheres in the BCC Model unit and the maximum achievable volume fraction is 68 vol.%, while there are four effective spheres in the FCC Model unit and the maximum achievable volume fraction is 74 vol.%. The established models are shown in Figure 2 and Figure 3.

#### 2.2.3. SA Model

The particles in Staggered Arrangement Model (SA Model) were arranged along the direction of particle incidence. The periodic repetition unit was built by U CARD, FILL CARD, TRCL CARD and LIKE BUT CARD in the MCNP5 code and input files were written using MATLAB. The building of repetition unit was mainly divided into two steps: (1) U CARD and FILL CARD were used to build the first and the second staggered layer respectively; (2) TRCL CARD and LIKE BUT CARD were used to copy and move these two layers until the shield was completely filled. The volume fraction of the particle was also achieved by controlling the side length of the unit. The side length of the unit was calculated according to the following formula:(3)lr=(43πr3/per)13
where r is the radius of spherical particles; per is the volume fraction of particles. 

The maximum achievable volume fraction was 54%, and the SA Model is shown in Figure 4.

### 2.3. The Whole Calculation Model

In this paper, neutron transmittance was calculated for B_4_C reinforced 316 stainless steel matrix composites and polyethylene matrix composites and the neutron flux distributions in 316 stainless steel were also analyzed. In these composites, B_4_C particles were arranged according to above four models. The entire calculation model is shown in Figure 5.

The radiation source was set as a plane source of neutrons of different energies (0.0253 eV, 50 eV, 50 keV, fission spectrum, ^241^Am-Be spectrum and 14.1 MeV) and the direction of neutron emission was perpendicular to the front surface of the shield. Among them, 0.0253 eV is the energy of thermal neutron, 14.1 MeV is the neutron energy of D-T fusion, the ^241^Am-Be spectrum will be compared with the experiment results and 50 eV and 50 keV represent resonance neutron and intermediate neutron, respectively.

Due to the wide range of neutron energies, it is unrealistic to adopt the same shield size and the thickness of the shield varied with neutron energies. Meanwhile, the size of the plane source and B_4_C particles changed with the size of the shield, as shown in Table 3. When the neutron is in the ^241^Am-Be spectrum, in order to be consistent with reference [22], the thickness of the 316 stainless steel composite was set to 10.0 cm. 

The F2 Tally Card allows users to calculate surface flux and the FMESH card allows the user to define a mesh tally superimposed over the problem geometry in the MCNP5 Program [23]. 

The F2 Card was applied to record the total neutron flux on the front surface of the tally and the statistical error was about 0.2–1% by controlling the number of the source neutrons. Then, neutron transmittance was calculated according to following formula:(4)$$T=\frac{\emptyset_{i}}{\emptyset_{0}}$$
where T is neutron transmittance; $\emptyset_{i}$ is the neutron flux when there is the shield; $\;\emptyset_{0}$ is the neutron flux when there is no shield.

The FMESH CARD was used to calculate the neutron flux distribution at surface 1 and surface 2, shown in Figure 5, and then neutron flux distribution images were drawn by MATLAB according to the output data. Surface 1 and surface 2 were divided into 100 × 100 equal parts.

## 3. Results and Discussion

### 3.1. Neutron Transmittance Results

The neutron transmittance of 1 vol.% to 30 vol.% B_4_C-reinforced 316 stainless steel composites and PE composites were calculated under four different models, respectively. The results are shown in Figure 6 and Figure 7.

It can be seen from Figure 6 and Figure 7 that, in general, for the two typical matrix composites reinforced by B_4_C, neutron transmittance under the four models is different when neutron energy is the same. That is to say that the spatial arrangement of B_4_C has an impact on the neutron shielding performance of the composites. Generally speaking, no matter what kind of matrix materials, the neutron shielding property of the RSA model composites is roughly better than the other three periodic arrangement composites except some energy neutrons, as shown in Figure 6f and Figure 7c. The results can be explained later according to the results of the neutron flux’s distribution. For the three periodic arrangement models, the neutron shielding performance of composites is related to B_4_C content. For example, when the neutron energy is 0.0253 eV, as shown in Figure 6a and Figure 7a, and when the volume fraction of B_4_C is less than 20 vol.%, the neutron transmittance of the SA model composite is smaller than that of the other two models, while the volume fraction of B_4_C is more than 20 vol.%, the neutron transmittance of the FCC composite is smaller. In addition, when the neutron is in a fission spectrum, as shown in Figure 6d and Figure 7d, the results of 316 stainless steel composites are consistent with the results of the 0.0253 eV neutron. However, for PE composites, the neutron transmittance of the SA model composite is the smallest.

At the same time, it is worth noting that the influence of B_4_C spatial arrangement on the shielding performance of the composites changes with neutron energies. The absorption cross section of ^10^B nuclides to thermal neutron is about three orders of magnitude higher than that of other nuclides in the composites, as shown in Table 2; thus, the different spatial arrangements of ^10^B can cause a difference in shielding performance. When neutron energy is 0.0253 eV, the neutron shielding performance of the four kinds of composites varies greatly. When the volume fraction of B_4_C is 30 vol.%, for 316 stainless steel composites, as shown in Figure 6a, the neutron transmittance of the FCC model composite is 13.35%, while that of the RSA model composite is only 1.96% and which is about 85.31% smaller than the former. For PE composites, the neutron transmittance of the FCC model composite is 10.64%, while that of the RSA composite is only 1.84%, about 82.71% smaller than the former, as shown in Figure 7a. However, when the neutron is fission spectrum and the content of B_4_C is also 30 vol.%, for 316 stainless steel composites, as shown in Figure 6d, the neutron transmittance of the FCC model composite is lower among three periodic models, which is 1.41%, while that of the RSA model composite is 1.36%, which is about 3.54% smaller than the former. Moreover, for PE composites, the neutron transmittance of the SA model composite is 0.76%, while that of the RSA model composite is 0.74%, which is about 2.63% smaller than the former, as shown in Figure 7d. Obviously, the difference of the shielding performance caused by B_4_C spatial arrangement when the neutron energy is 0.0253 eV and it is much larger than when evaluated with the fission spectrum.

For 316 stainless steel composites, by comparing 50 eV and 50 keV, as shown in Figure 6b,c, more thermal neutrons will be generated in the former case after slowing down because the microscopic cross section of ^10^B nuclide is inversely proportional to the neutron’s energy [27], so the shielding performance difference caused by spatial arrangement is relatively obvious. At this point, when the volume fraction of B_4_C is 30 vol.% and the neutron transmittance of the FCC model composite is 25.65%, while that of the RSA model composite is 24.66%, which is about 3.86% smaller than the former. For the fission neutron spectrum, its energy range is wide. In addition to thermal neutrons in source spectrum, thermal neutrons can also be generated by slowing. However, the proportion of thermal neutron is low; thus, the difference of the shielding performance caused by different spatial arrangements is not so obvious. As mentioned above, the neutron transmittance of the RSA model composite is 3.54% smaller than the FCC model composite. When the neutron energy is 14.1 MeV, as shown in Figure 6f, the inelastic scattering effect of the Fe element is better than B_4_C and the proportion of moderated thermal neutron is limited because the energy of neutron is so high, so the shielding property of the composites improved by B_4_C is not clear, and the influence of the spatial arrangement is extremely small, which is equivalent to the statistical error.

For PE composites, when neutron energy is 50 eV and the fission spectrum is used, the results are similar to those of 316 stainless steel composites, as shown in Figure 7b,d. For fission spectrum neutrons, when the volume fraction of B_4_C is 30 vol.%, the neutron transmittance of the SA model composite is 0.76%, while that of the RSA model composite is 0.74%, which is about 2.63% smaller than the former. Moreover, for 50 keV neutrons, as shown in Figure 7c, due to the fact that the elastic scattering of the H element is better than B_4_C and the slow down of the neutron thermal is extremely limited, the influence of the spatial arrangement is extremely small, which is equivalent to the statistical error. At the same time, it can be seen that with the increase in B_4_C, neutron transmittance tends to increase; that is, the neutron shielding performance of the composites decreases because the elastic scattering effect of matrix PE material is weakened. When neutron energy is 14.1 MeV, as shown in Figure 6f and Figure 7f, the neutron transmittance of PE composites is larger than that of 316 stainless steel composites, which is mainly because the inelastic scattering effect of Fe element is better than that of H and C elements. At this point, the spatial arrangement had a little effect on the neutron shielding performance of the composites.

In addition, when the neutron is the ^241^Am-Be spectrum, for 316 stainless steel composites, as shown in Figure 6e, the neutron transmittance of the RSA model composite is lowest, and the difference is within 1.5% of the other three periodic models. In reference, it can be found that there is a little difference between the BCC simulation results and the experimental results of the B_4_C randomly distributed composites. When the thickness of the B_4_C-reinforced 304 stainless steel is 10 cm, the BCC simulation result is about 2.84% lower than the experimental results [22]. Combining these two results, it can be concluded that the simulation results of the RSA Model and the experiment results are both lower than the simulation results of periodic models. This also means that the RSA Model matched the experiment better than periodic models. For PE composites, the neutron transmittance of the FCC model composite is the lowest and the difference is equivalent to the statistical error. In general, it can be shown that there is little difference between RSA model and periodic models under the ^241^Am-Be spectrum, and the RSA Model can be replaced by periodic models to evaluate the neutron performance of the composites for the ^241^Am-Be spectrum.

### 3.2. Neutron Flux Distribution Results

As surface 1 and surface 2 show in Figure 5, on the *x-z* plane, the neutron flux distributions in the shield were calculated and the neutron flux distributions are shown in Figure 8 and Figure 9. The shield is 10 vol.% B_4_C-reinforced 316 stainless steel composites. The source is 0.0253 eV neutrons in Figure 8 and fission spectrum neutrons in Figure 9. Moreover, the maximum and the minimum relative neutron fluxes are listed in Table 3 and the statistical error was about 0.1–2.0%.

As shown in Figure 8, for thermal neutrons, the neutron absorption cross section of ^10^B is much higher than that of 316 stainless steel. Therefore, it can be considered that thermal neutrons will be absorbed in regions with B_4_C distribution, while ‘neutron channels’ will be formed in regions without B_4_C distribution. Therefore, there are obvious differences in neutron flux distribution under different spatial arrangements. At surface 1, the neutron flux distributions of all models present two regions of alternating intensity, namely the bright region and the dark region. As shown in Table 4, the maximum neutron flux of the brightest region under four models is basically the same. At the same time, the minimum neutron flux of the darkest region under RSA model is similar to that of the BCC model and shows no obvious advantage. However, compared with surface 1, the bright area at surface 2 under RSA model decreases significantly and the dark area increases, while the bright area and dark area at surface 2 under three periodic arrangement models remain unchanged compared with surface 1. This is mainly because the position of B_4_C is randomly distributed under the RSA model, and the effect of ‘alternating shielding’ will be formed between B_4_C particles. At surface 1, due to the small number of B_4_C particles at the front, the ‘alternating shielding’ effect was poor, while at surface 2, due to the increased number of B_4_C particles at the front, the effect of ‘alternating shielding’ is more obvious, which increases the probability of thermal neutron absorption. However, for the three periodic arrangement models, B_4_C particles are always regularly arranged and the ‘neutron channels’ are formed in the fixed direction and there is no ‘alternating shielding’ effect; thus, the bright area and the dark area keep unchanged. Meanwhile, it can be seen from Table 4 that, at surface 2, the minimum neutron flux of the brightest area under RSA model is significantly smaller than that under the three periodic arrangement models, which is about 25.20% smaller than that under the SA model while the statistical error is about 2.0%. This is consistent with the rule shown in Figure 6a where the neutron transmittance of the RSA model composite is much smaller than that of the three periodic arrangement models composites when the neutron is 0.0253 eV. 

In addition, as shown in Figure 6a, the difference of neutron transmittance between RSA model and the three periodic arrangement models increases first and then decreases with the increase in B_4_C volume fraction, which can also be explained by ‘neutron channels’ and ‘alternating shielding’: When the volume fraction of B_4_C is small, there are many of ‘neutron channels’ either in the RSA model or in three periodic arrangement models and the ‘alternating shielding’ effect cannot be formed effectively; thus, the difference between them is small. However, with the increase in B_4_C content, such as 10 vol.%, the effective ‘alternating shielding’ can be formed in the RSA model, while some ‘neutron channels’ still exist in three periodic arrangement models, as shown in Figure 8; thus, the neutron transmittance difference between them is much larger. It can be observed from Figure 2, Figure 3 and Figure 4 that as the volume fraction of B_4_C further increases, the spacing between B_4_C particles in the periodic arrangement models becomes smaller and smaller; that is, the area of ‘neutron channels’ decreases and a certain degree of ‘alternating shielding’ can be achieved. Therefore, the neutron transmittance difference between the RSA model and the periodic arrangement models decreases.

As shown in Figure 9, for fission neutrons, under different spatial arrangements, circular spots with high neutron flux at the central position and low surrounding neutron flux are presented. The average energy of the fission spectrum is 1.98 MeV and the scattering effect is the main interaction. Therefore, the neutrons will converge to the center and the circular spots are formed. At surface 1, the proportion of thermal neutrons is not high, so there are no obvious alternating regions of bright and dark, as shown in Figure 8. At surface 2, the proportion of thermal neutrons is relatively higher after slowing down, so a more obvious bright-dark alternating region appears compared with surface 1 under the three periodic models. In general, at surface 2, although the neutron flux intensity at the brightest region under the RSA model is smaller than that of the other three periodic arrangement models, as shown in Table 4, the difference is not significant and is about 3.18% smaller than that of the SA model while the statistical error is about 0.1%, which is also consistent with the results in Figure 6d.

## 4. Conclusions

In this paper, a functional particle random arrangement model and three periodic arrangement models were established: RSA Model, FCC Model, BCC Model and SA Model. The neutron transmittances of 316 stainless steel composites and PE composites reinforced by B_4_C particles were calculated under different energy neutrons to compare the shielding performance. In addition, the neutron flux distributions of 316 stainless steel composites under 0.0253 eV and fission spectrum neutrons were also analyzed separately. Combined with the microscopic cross section of B, C, H and Fe elements, the ‘alternating shielding’ and the ‘neutron channels’ were presented to explain the reason why the spatial arrangement of B_4_C has an impact on the neutron shielding performance of the composites. Several main conclusions are provided,

(a)The spatial arrangement of B_4_C has an impact on the neutron shielding performance of the composites. The RSA model composites are roughly better than the other three periodic arrangement composites.(b)The influence of B_4_C spatial arrangement on the shielding performance of the composites changes with neutron energies. When the neutron energy is 0.0253 eV and the B_4_C is 30 vol.%, for 316 stainless steel composites, the neutron transmittance of the RSA model composite is about 85.31% smaller than the FCC model composite, while for the PE composite, it is about 82.71% at the same case. However, When the neutron is in the fission spectrum and the content of B_4_C is 30 vol.%, for 316 stainless steel composites, the neutron transmittance of the RSA model composite is about 3.54% smaller than the FCC model composite, while for PE composite, it is only about 2.63%.(c)For the three periodic arrangement models, the neutron shielding performance of the composites is related to B_4_C content. In the case of 0.0253 eV neutrons, when B_4_C is less than 20 vol.%, the neutron transmittance of the SA model composite is smaller than that of the other two models; while B_4_C is greater than 20 vol.%, the neutron transmittance of the FCC model composite becomes smallest. In the case of fission neutron, the results of 316 stainless steel composites are consistent with the results of 0.0253 eV neutrons. However, for PE composites, the neutron transmittance of the SA model composite is basically smallest.(d)The more effective alternating shielding is, the fewer ‘neutron channels’ and the better the neutron shielding performance of the composites.

It can be concluded that the selection of shielding calculation models should be conducted with more care because of the influence of reinforced particles and their spatial arrangements on the neutron shielding performance of the composites. When it is at the micron grade, B_4_C- or B-reinforced composites and the proportion of thermal neutrons are not small, and the RSA model is much more accurate and more suitable. Moreover, when it is the millimeter grade of B_4_C- or B-reinforced composites and the source is a fission spectrum neutron or ^241^Am-Be spectrum, periodic arrangement models can be applied for materials optimized design and shielding performance calculation because of the small difference between the RSA Model composite and the periodic models composites and their shorter calculation time and higher calculation efficiency.

## Figures and Tables

**Figure 1 materials-15-04266-f001:**
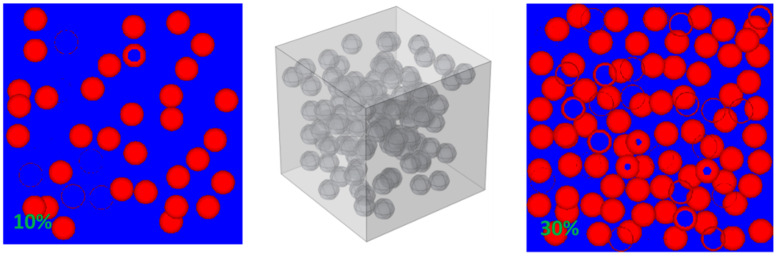
Two-dimensional and three-dimensional RSA Model (10 vol.% on the left and 30 vol.% on the right).

**Figure 2 materials-15-04266-f002:**
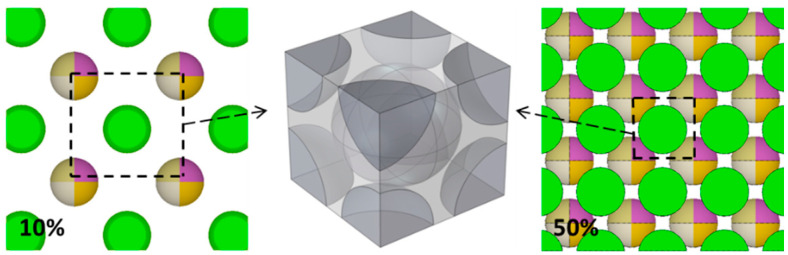
Two-dimensional BCC Model (10 vol.% on the left and 50 vol.% on the right) and three-dimensional BCC Model unit.

**Figure 3 materials-15-04266-f003:**
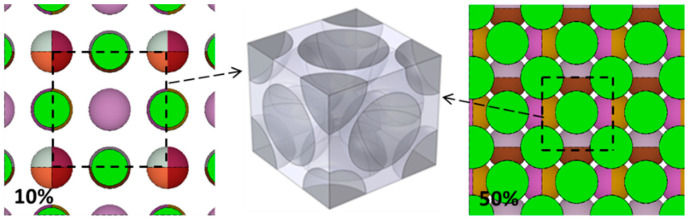
Two-dimensional FCC Model (10 vol.% on the left and 50 vol.% on the right) and three-dimensional FCC Model unit.

**Figure 4 materials-15-04266-f004:**
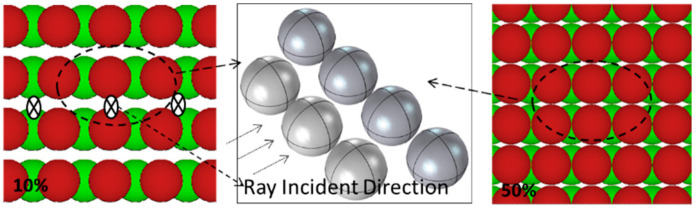
Two-dimensional SA Model (10 vol.% on the left and 50 vol.% on the right) and three-dimensional SA Model unit.

**Figure 5 materials-15-04266-f005:**
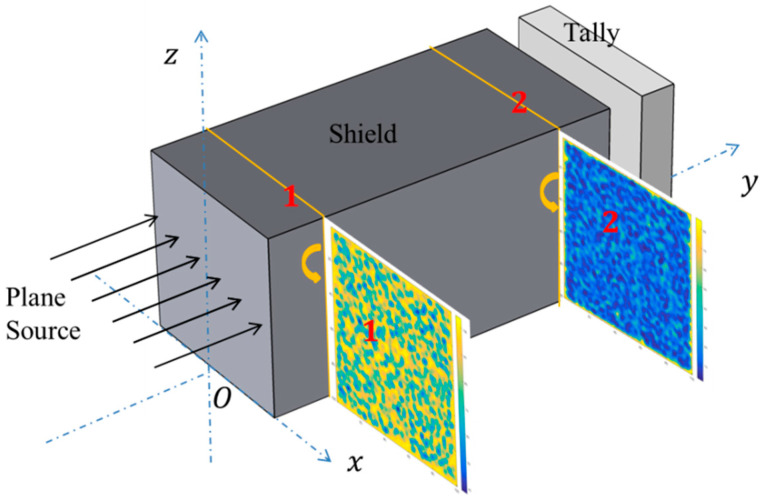
The entire calculation model (surface 1 and surface 2 are on the *x-z* plane for neutron flux distribution).

**Figure 6 materials-15-04266-f006:**
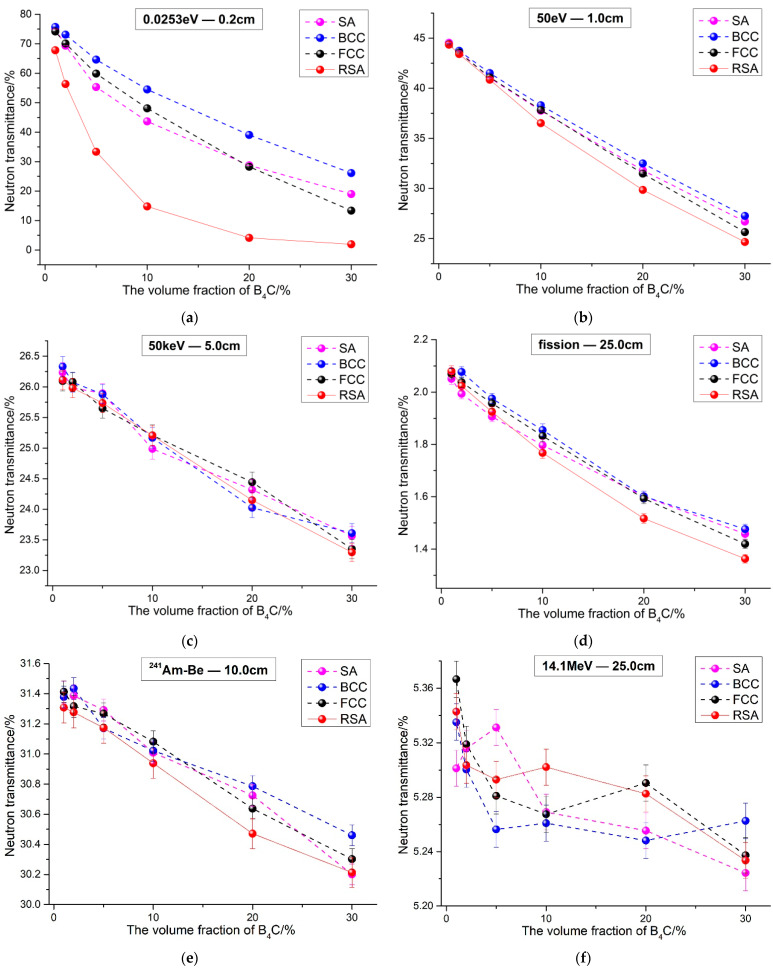
Neutron transmittance of B_4_C reinforced 316 stainless steel composites under different neutron energies. ((**a**) 0.0253 eV; (**b**) 50 eV; (**c**) 50 keV; (**d**) fission spectrum; (**e**) ^241^Am-Be spectrum; (**f**) 14.1 MeV).

**Figure 7 materials-15-04266-f007:**
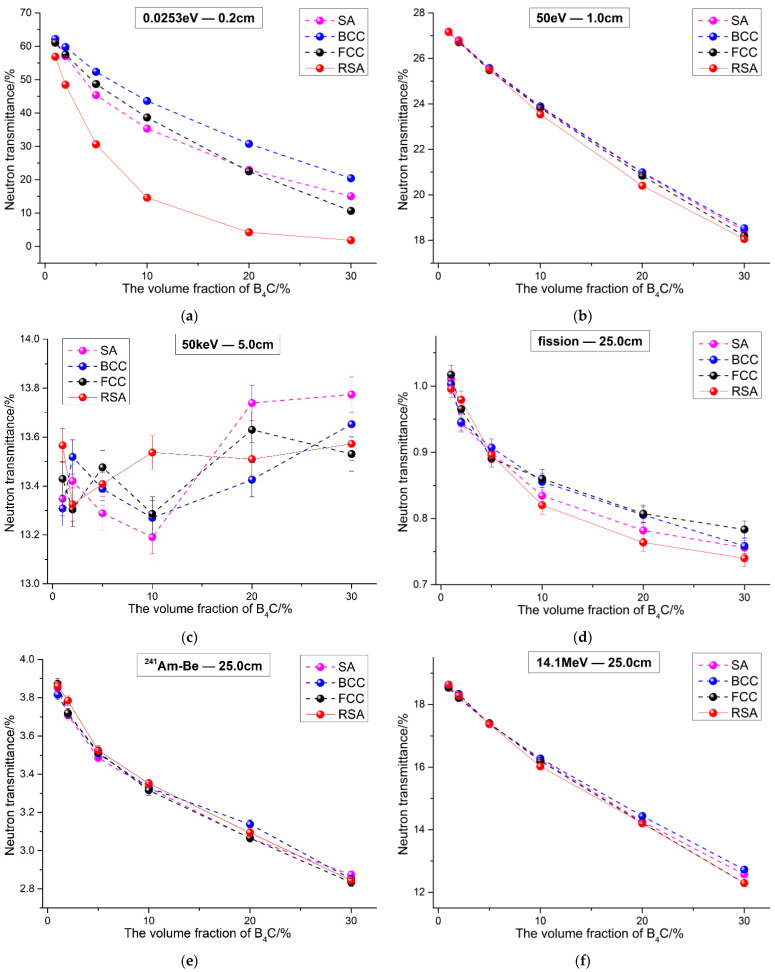
Neutron transmittance of B_4_C reinforced PE composite under different neutron energies. ((**a**) 0.0253 eV; (**b**) 50 eV; (**c**) 50 keV; (**d**) fission spectrum; (**e**) ^241^Am-Be spectrum; (**f**) 14.1 MeV).

**Figure 8 materials-15-04266-f008:**
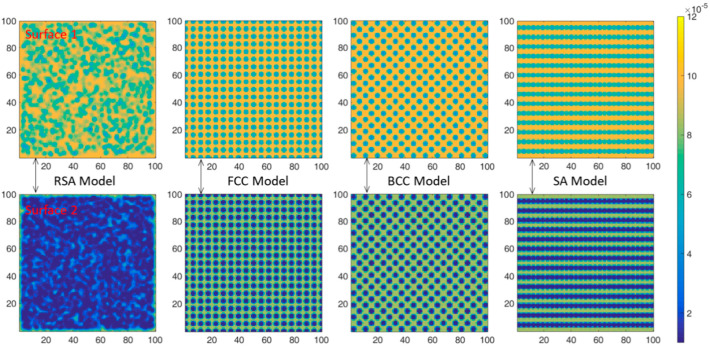
In the case of 0.0253 eV neutrons, the neutron flux distributions at surface 1 and surface 2 under different spatial arrangements.

**Figure 9 materials-15-04266-f009:**
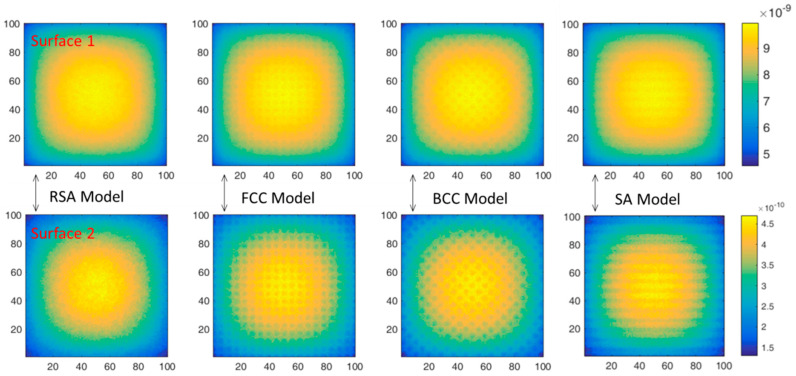
In the case of fission spectrum, the neutron flux distributions at surface 1 and surface 2 under different spatial arrangements.

**Table 1 materials-15-04266-t001:** The composition and density of materials.

Elements	H	^10^B	^11^B	C	Si	P	S	Cr	Mn	Fe	Ni	Mo	Density(g·cm^−3^)
Materials	Content (wt%)												
B_4_C	-	15.7	62.6	21.7	-	-	-	-	-	-	-	-	2.52
316 stainless steel	-	-	-	0.08	1.00	0.035	0.03	17.75	2	63.61	13	2.5	7.62
PE	14.3	-	-	85.7			-	-	-	-	-	-	0.95
1 vol.% B_4_C/316 stainless steel	-	0.05	0.21	0.15	1.00	0.035	0.030	17.69	1.99	63.40	12.96	2.49	7.57
5 vol.% B_4_C/316 stainless steel	-	0.27	1.07	0.45	0.98	0.034	0.029	17.45	1.97	62.52	12.78	2.46	7.37
10 vol.% B_4_C/316 stainless steel	-	0.56	2.22	0.78	0.96	0.034	0.029	17.12	1.93	61.36	12.54	2.41	7.11
20 vol.% B_4_C/316 stainless steel	-	1.20	4.78	1.73	0.92	0.032	0.028	16.39	1.85	58.75	12.01	2.31	6.60
30 vol.% B_4_C/316 stainless steel	-	1.95	7.77	2.76	0.88	0.031	0.026	15.55	1.75	55.71	11.39	2.19	6.09
1 vol.% B_4_C/PE	13.93	0.41	1.63	84.03	-	-	-	-	-	-	-	-	0.97
5 vol.% B_4_C/PE	12.55	1.92	7.67	77.86	-	-	-	-	-	-	-	-	1.03
10 vol.% B_4_C/PE	11.04	3.57	14.25	71.13	-	-	-	-	-	-	-	-	1.11
20 vol.% B_4_C/PE	8.60	6.26	24.96	60.18	-	-	-	-	-	-	-	-	1.26
30 vol.% B_4_C/PE	6.69	8.35	33.30	51.65	-	-	-	-	-	-	-	-	1.42

**Table 2 materials-15-04266-t002:** The neutron microscopic total cross sections of main nuclides.

Energy (MeV)	Nuclides				
	^1^H	^10^B	^12^C	^52^Cr	^56^Fe
2.53 × 10^−8^	30.40	3846.76	4.95	3.93	14.78
5 × 10^−5^	20.44	88.30	4.75	3.06	11.96
5 × 10^−2^	15.53	5.14	4.59	36.46	4.23
Fission (average: 1.98)	2.91	2.06	1.70	2.73	2.38
^241^Am-Be (average: 4.5)	1.75	1.79	1.69	3.81	3.80
14.1	0.69	1.49	1.30	2.41	2.59

**Table 3 materials-15-04266-t003:** The size of the shield, spherical B_4_C and the plane source.

Energy (MeV)	The Shield Size (cm)	The B_4_C Radius (cm)	The Plane Source Size (cm)
2.53 × 10^−8^	0.1 × 0.2 × 0.1	0.002	0.1 × 0.1
5 × 10^−5^	0.5 × 1.0 × 0.5	0.01	0.5 × 0.5
0.05	2.5 × 5.0 × 2.5	0.05	2.5 × 2.5
Fission (average: 1.98)	12.5 × 25.0 × 12.5	0.25	12.5 × 12.5
^241^Am-Be (average: 4.5)	12.5 × 10.0/25.0 × 12.5	0.25	12.5 × 12.5
14.1	12.5 × 25.0 × 12.5	0.25	12.5 × 12.5

**Table 4 materials-15-04266-t004:** The maximum and minimum relative neutron flux of surface 1 and surface 2 under different conditions.

Position	Neutron Flux	0.0253 eV				Fission Spectrum			
		RSA	FCC	BCC	SA	RSA	FCC	BCC	SA
Surface 1	maximum	1.04 × 10^−4^	1.03 × 10^−4^	1.04 × 10^−4^	1.03 × 10^−4^	9.96 × 10^−9^	9.93 × 10^−9^	9.94 × 10^−9^	9.95 × 10^−9^
	minimum	3.77 × 10^−5^	4.25 × 10^−5^	3.78 × 10^−5^	4.71 × 10^−5^	5.79 × 10^−9^	5.75 × 10^−9^	5.77 × 10^−9^	5.68 × 10^−9^
Surface 2	maximum	6.50 × 10^−5^	8.65 × 10^−5^	8.61 × 10^−5^	8.69 × 10^−5^	4.56 × 10^−10^	4.71 × 10^−10^	4.75 × 10^−10^	4.71 × 10^−10^
	minimum	1.20 × 10^−6^	1.12 × 10^−6^	1.11 × 10^−6^	1.41 × 10^−6^	1.65 × 10^−10^	1.61 × 10^−10^	1.61 × 10^−10^	1.60 × 10^−10^

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
