# Peer review of "Study on the Influence of Reinforced Particles Spatial Arrangement on the Neutron Shielding Performance of the Composites"

_materials, 2022, doi:10.3390/ma15124266_

Round 1

Reviewer 1 Report

This manuscript is studied for the calculation of neutron shielding performance with B4C, Stainless steel and composition. English presentation and theoretical framework are good. However, in this study, the author ignores material science and crystallography. Any assumed compose materials can be used easily in the calculation. However, realization of material for shielding is the one of important items. Therefore, enough consideration and enough discussion in the viewpoint of material science and crystallography should be done. The reviewer recommends that this manuscript should be rejected.

Minor comment

In Sec.2: How about weight density or number density of composited materials? When the B4C volume is changed, how is the density changed?

In Sec.2.3: Is direction of the radiation source perpendicular to the plane? or not?

In Sec.2.3: Are incident neutron energies mono except the fission spectrum?

In Sec.2.3: The reason to shield size change cannot be understood.

In Sec.2.3: How about energy change and direction change of after the penetration of the shielding?

In Sec.2.3: How about gamma-ray after the penetration of the shielding?

At 4th line in p.4: The function of MATLAB is not clear. And the reviewer should cite MATLAB.

In table 2: Do these values mean total cross section?

In table 4: Should show the error value.

In Fig.6: In the 14.1 MeV data, the statistical error is so large. The author should decrease .

Several references are cited with no page and no volume.

Reviewer 2 Report

  • In the introduction, you can add a few more sentences to the last paragraph about the extent of the study.
  • Not enough background and literature provided. The introduction should tackle the problem and introduce the reason why this study was carried out and provide a clear hypothesis regarding the problem statement.
  • The author should justify the energy used in this study according to their applications.
  • Please justify choosing these samples for the purpose of radiation protection

Reviewer 3 Report

Dear all,
as a general comment, I understood that the authors compared different ways of spacial arrangement for the particles in composite materials and they show that the shielding performance is affected by how you model the arrangement of the particles however there is no comparison with any real data that leave different to decide which is the right model to choose for a simulation I would suggest that the authors add a description how the 4 models compared to existing experimental data

I think the overall English quality of the paper should be improved there are a lot of mistakes like plurals
for example
"The particle reinforced composites are widely applied as nuclear radiation shielding 19 material" should be replaced with
"The particle reinforced composites are widely applied as nuclear radiation shielding 19 materials"
"calculated under different energy neutrons source"
with "calculated under different energy neutrons sources" and there are many of these in the text. There are also parts of the texts where the content is not fully clear for example line 65 W size should be replaced with tungsten.
There are also parts of the text where the content is not fully clear for example
line 82 "is much longer than the periodic models" here the content is not clear they want to say that the calculations take longer but it should be written explicitly

Some of the references are also very difficult to access for reference 1 for example there is not enough information to find the paper they are referring to.

It is also never mentioned with version of the MCNP code has been used for the calculations
They should also write clear which kind of library are they using

In table 2 the different materials should be separated by a line to make it easier to distinguish between each other

When they mention the U CARD and all the other cards in mcnp they should specify the MCNP version and reference the manual

the layout of Table 3 should also be improved Energy (MeV) radius (cm) ecc..

Are the errors only statistical coming from the Monte Carlo? >Did they estimate the systematic errors of the simulations? Usually, people are doing that by changing the library versions or the nuclear model

Round 2

Reviewer 1 Report

This manuscript is studied for the calculation of neutron shielding performance with B4C, stainless steel and composition. English presentation and theoretical framework are good. The author well revised the manuscript. The manuscript is proper as a paper as the journal. However, as there are no space between a value and a unit all over in the manuscript, the reviewer recommends the minor revision.

Author Response

Dear reviewer,

    Thanks for your professional and helpful comments.

    The problem you mentioned has been corrected in our revision manuscript.

    Some other corrections have also been done in our revision manuscript.

                                                                                              Yours, Prof. Huasi Hu

Reviewer 2 Report

Accept

Author Response

Dear reviewer,

    Thanks for your professional and helpful comments.

                                                                 Yours, Prof. Huasi Hu